# Prevalence of Unfavorable Video-Urodynamic Findings and Clinical Implications in Patients with Minimally Conscious State/Unresponsive Wakefulness Syndrome: A Retrospective Descriptive Analysis

**DOI:** 10.3390/biomedicines11092432

**Published:** 2023-08-31

**Authors:** Francois Leboutte, Christian Engesser, Leutrim Zahiti, Cyrill A. Rentsch, Helge Seifert, Ralf Anding, Margret Hund-Georgiadis, Sandra Möhr, Matthias Walter

**Affiliations:** 1Department of Urology, University Hospital Basel, University of Basel, 4031 Basel, Switzerland; lebouttefr@hotmail.com (F.L.); christian.engesser@usb.ch (C.E.); l.zahiti@gmx.ch (L.Z.); cyrill.rentsch@usb.ch (C.A.R.); helge.seifert@usb.ch (H.S.);; 2Clinic of Neurorehabilitation and Paraplegiology, REHAB Basel, 4055 Basel, Switzerland; m.hund@rehab.ch; 3Neuro-Urology, REHAB Basel, 4055 Basel, Switzerland

**Keywords:** bladder-emptying methods, detrusor sphincter dyssynergia, indwelling catheters, minimally conscious state, neurogenic detrusor overactivity, neurogenic lower urinary tract dysfunction, treatment, unresponsive wakefulness syndrome, video-urodynamic studies

## Abstract

The aim of this retrospective exploratory study was to investigate the prevalence of unfavorable findings during video-urodynamic studies (VUDS) in patients with minimally conscious state (MCS)/unresponsive wakefulness syndrome (UWS) and whether management of the lower urinary tract (LUT) was adjusted accordingly. A retrospective chart review was conducted to screen for patients diagnosed with MCS/UWS at our rehabilitation center between 2011 and 2020. Patients 18 years or older were included and underwent baseline VUDS after being diagnosed with MCS/UWS. We analyzed urodynamic parameters and subsequent changes in LUT management in this cohort. In total, 32 patients (7 females, 25 males, median age 37 years) with MCS/UWS were included for analysis. While at least one unfavorable VUDS finding (i.e., neurogenic detrusor overactivity [NDO], detrusor sphincter dyssynergia {DSD, high maximum detrusor pressure during storage phase [>40 cmH_2_O], low-compliance bladder [<20 mL/cmH_2_O], and vesico–uretero–renal reflux [VUR]) was found in each patient, NDO (78.1%, 25/32) and DSD (68.8%, 22/32) were the two most frequent unfavorable VUDS findings. Following baseline VUDS, new LUT treatment options were established in 56.3% (18/32) of all patients. In addition, bladder-emptying methods were changed in 46.9% (15/32) of all patients, resulting in fewer patients relying on indwelling catheters. Our retrospective exploratory study revealed a high prevalence of NDO and DSD in patients with MCS/UWS, illustrating the importance of VUDS to adapt LUT management in this cohort accordingly.

## 1. Introduction

Initially coined as a “vegetative state” (VS) by Jennet and Plum in 1972 [1], the unresponsive wakefulness syndrome (UWS), as proposed by the European Task Force on Disorders of Consciousness in 2010, “refers to patients showing several clinical signs (hence syndrome) of unresponsiveness (that is, without response to commands) in the presence of wakefulness (that is, eye opening)” [2]. While the American Academy of Neurology [3] refers to this disorder of consciousness as VS/UWS, the Royal College of Physicians [4] decided to remain with the term VS. In contrast, patients with a minimally conscious state (MCS) are characterized by “the presence of minimal but clearly discernible behavioral evidence of self or environmental awareness”. Further, MCS was defined by the Aspen Neurobehavioral Conference Workgroup in 2002 as “a condition of severely altered consciousness in which minimal but clearly discernible behavioral evidence of self or environmental awareness is demonstrated” [3]. While UWS and MCS are both hallmarked by severe impairments of consciousness and associated with a substantial social and economic burden [5], MCS differs considerably from VS/UWS, which is supported by recent evidence from neurobehavioral and neuroimaging studies [6]. However, beneficial evidence of neurological rehabilitation in MCS and VS/UWS has been demonstrated previously [7,8,9]. Furthermore, early rehabilitation has been reported to be associated with better outcomes in severely traumatic brain-injured patients [10,11].

However, little is known about the impact of MCS and VS/UWS on a patient’s lower urinary tract (LUT) function, in particular on urine storage and voiding. To our knowledge, only a few studies have conducted urodynamic studies (UDS) to assess LUT function in patients with VS or MCS/UWS [12,13,14,15,16]. In line with other neurological disorders, LUT function should be assessed and aided appropriately to protect or at least minimize the risk of structural and functional changes of the upper urinary tract (UUT) and LUT in the future [17]. Well-known complications and long-term consequences include recurrent urinary tract infection, stone formation (urinary bladder and kidney), and vesico–uretero–renal reflux (VUR), as well as renal insufficiency and renal failure [18].

Although patients with MCS/UWS can be assessed clinically, verbal assessments are mostly impossible given the patients’ impaired cognitive function. Complaints such as “I am wetting” or “I feel a burning sensation” cannot be expected from any individual in this cohort. However, objective assessments, such as sonography, cystoscopy, and UDS, can be performed to evaluate LUT and UUT. However, UDS is the only method to objectively assess LUT function (level of evidence [LE] 2a) [19], and Video-UDS (VUDS) is considered the preferred option to conduct UDS in individuals with underlying neurological disorders (LE 4) [19]. However, VUDS are not routinely performed in clinical practice.

Owing to the lack of evidence, we aimed to identify unfavorable VUDS findings in patients with MCS/UWS and whether clinical treatment was adjusted accordingly to improve LUT function in this cohort.

## 2. Materials and Methods

### 2.1. Population

Patients were eligible when meeting the following inclusion criteria: aged 18 years or older, diagnosis of MCS or UWS at our rehabilitation center between 2011 and 2020, and documented baseline VUDS for assessment of LUT function. No limits were set in terms of the time point of baseline VUDS following MCS/UWS diagnosis. Further, patients must not have had any spinal fractures or spinal cord injury (SCI), which were accounted for using computed tomography or magnetic resonance imaging.

### 2.2. Objectives

The primary outcome was to obtain the prevalence of unfavorable urodynamic findings [20] (i.e., high maximum detrusor pressure [Pdet] during the storage phase [>40 cmH_2_O], low-compliance bladder [<20 mL/cmH_2_O], neurogenic detrusor overactivity (NDO), detrusor sphincter dyssynergia (DSD), and VUR) during VUDS in patients with MCS/UWS. The secondary outcome was to compare patients who underwent VUDS “Early” (i.e., ≤12 months) versus “Late” (i.e., ≥13 months) after diagnosis and whether LUT management was changed with respect to VUDS findings.

### 2.3. Outcome Variables

We chose the following categorical outcome variables: sex (female and male), type of underlying brain injury resulting in MCS/UWS (traumatic, hypoxic, or hemorrhagic), presence of NDO (yes or no), presence of Pdetmax during storage ([>40 cmH_2_O], yes or no), presence of low-compliance bladder during filling cystometry ([<20 mL/cmH_2_O] yes or no), presence of DSD (yes or no), presence of VUR (yes or no), method of bladder emptying (indwelling catheter (transurethral/suprapubic), condom catheter, intermittent catheterization (i.e., self or assisted), or volitional voiding), and specific pharmaceutical LUT therapies, i.e., antimuscarinics (yes or no), alpha-blockers (yes or no), mirabegron (yes or no), intrasphincteric botulinum toxin-A injections (yes or no), intradetrusor botulinum toxin-A injections (yes or no).

We chose the following continuous outcome variables: age [years], time from diagnosis to VUDS (months), neurological/cognitive function prior to VUDS, i.e., coma recovery scale—revised (CRS-R) [21], Glasgow Coma Scale (GCS) [22], and/or Bavesta Score [23], VUDS parameters, i.e., maximum cystometric capacity (MCC, [mL]), Pdetmax during storage [cmH_2_O], bladder compliance during filling cystometry [mL/cmH_2_O], Pdetmax during voiding [cmH_2_O], and detrusor overactivity leak point pressure (DOLPP, [cmH_2_O]), detrusor overactivity leak volume (DOLV, [mL]), and post-void residual (PVR) urine [mL].

### 2.4. Data Source and Collection

Data were collected between 2011 and 2020 (Figure 1). We retrospectively obtained data from medical charts (i.e., categorical and continuous outcome variables). VUDS were conducted using the urodynamic system (Uromic Quickstep, Medkonsult Medical Technology, MMT; Olomouc, Czech Republic) in combination with the fluoroscopy system (Artis zee multi-purpose, Siemens Healthineers International AG; Erlangen, Germany). All urodynamic assessments were performed in accordance with the International Continence Society’s “Good Urodynamic Practices” [24] to evaluate LUT function and quantify the current extent of neurogenic lower urinary tract dysfunction (NLUTD).

A semi-quantitative assessment of the level of consciousness was performed using three different scales to establish diagnosis and monitor behavioral recovery. Thus, CRS-R [21] and GCS [22] are two internationally well-established scores to assess the level of consciousness. The BAVESTA score [23] is an interprofessional observation tool to measure convalescence in MCS/UWS, which uses relevant functions of patients suffering from severe consciousness impairment, such as gazing, muscle tone, and hygiene. Furthermore, it has been validated and is commonly used in our institution for follow-up in MCS/UWS [25].

### 2.5. Bias

We encountered two kinds of bias: (1) selection bias—all patients were retrieved retrospectively during a pre-defined 10-year period, and (2) temporal bias—the time point of VUDS across all patients was not standardized and, therefore, comprised a significant variance.

### 2.6. Statistical Methods

Statistical analyses were conducted using R (Version 4.0.5 for Mac Os). Non-parametric statistics were applied. A Wilcoxon signed rank test (i.e., age, time from diagnosis to VUDS, CRS-R, GCS, Bavesta score, Pdetmax storage, DOLPP, DOLV, Compliance, MCC and PVR), Fisher exact test (i.e., sex, NDO, Pdetmax storage > 40 cmH_2_O, low-compliance bladder and DSD), and Kruskal–Wallis Test (i.e., type of brain injury) were used to compare between “Early” and “Late” VUDS. Data are presented as raw values and percentages, medians with lower (Q1) and upper quartiles (Q3), as well as minimum and maximum, wherever indicated. The threshold for a statistically significant difference was *p* < 0.05.

## 3. Results

Overall, 32 patients (7 females, 25 males, median age 37 years, median time from diagnosis to VUDS 4 months) diagnosed with MCS/UWS were included in this study. Demographics and injury characteristics are highlighted in Table 1.

Prior to baseline VUDS, the minority of patients had at least one pharmaceutical LUT treatment (21.9%) and utilized one method of assisted urinary bladder emptying (78.1%) (see Table 2). Overall, pharmaceutical LUT treatment changed as a direct consequence of the VUDS in 56.3% (18/32) of patients (see Table 2).

Furthermore, 46.9% (15/32) of all patients had experienced at least one LUT complication since admission, comprising UTI/CA-UTI (defined as the presence of symptoms or signs compatible with UTI with no other identified source of infection along with 10^3^ colony-forming units) [26] in 31.3% (10/32) and catheter occlusion in 21.9% (7/32). While all patients demonstrated at least one unfavorable urodynamic parameter, NDO and DSD were the two most frequent (see Table 3).

## 4. Discussion

### 4.1. Main Findings

Our retrospective exploratory study revealed novel findings highlighting the extent of NLUTD in patients with MCS/UWS. We found a high prevalence in three of the five unfavorable urodynamic findings (i.e., NDO, Pdetmax during storage > 40 cmH_2_O, and DSD), which led to subsequent therapeutic changes in our cohort of 32 patients with MCS/UWS.

### 4.2. Findings of Unfavorable Urodynamic Parameters

At least one unfavorable urodynamic finding was detected in each patient of our cohort—most commonly NDO and DSD. A similar high prevalence of these two unfavorable urodynamic parameters has been recently reported in individuals with SCI [27].

NDO is highly prevalent (i.e., up to 95%) in individuals with suprasacral SCI after the initial spinal shock phase [28] and to a lesser degree in individuals with multiple sclerosis (i.e., ~43%) [29] or myelomeningocele (i.e., ~44%) [30]. In our cohort, NDO was detected in 78% of all patients, which is in line with Benecchi et al. [14], who reported a prevalence of 79% in a cohort of 20 patients with VS. In contrast, both Girando et al. [13] (i.e., 63%, 10/16) and Windaele [16] (i.e., 33%, 5/15) reported a lower prevalence of NDO in patients with MCS/UWS. In our cohort, 85% (21/25) of all patients with NDO had a Pdetmax during storage > 40 cmH_2_O.

DSD is highly prevalent (i.e., >70%) in individuals with suprasacral SCI after the initial spinal shock phase [31] and to a lesser degree in individuals with multiple sclerosis (i.e., ~36%) [29] or myelomeningocele (i.e., ~56%) [30]. In our cohort, the prevalence of DSD was close to 70%. This finding contrasts the reports from Krimchansky et al. [12], who could not find any DSD in a cohort of 17 patients in a post-traumatic VS, as well as from Windaele [16], who reported only one of fifteen patients with DSD, in whom a spinal cord lesion was discovered in the course of the disease. Thus, none of the patients with VS/coma due to brain injury presented with DSD. Further, neither Benecchi et al. [14] nor Girando et al. [13] reported on the presence of DSD. Considering that DSD, defined as a detrusor contraction concurrent with an involuntary contraction of the urethral and/or periurethral striated muscle [32], is a consequence of disruption of central nervous system regulation of the micturition reflex and is usually seen in patients with SCI. In our cohort, neuro-imaging was used to screen for SCI. However, none of the 32 patients presented a SCI, which raises the question of what the underlying mechanism for DSD in our cohort might be.

While none of the other studies reported on the presence of VUR, in our cohort, fortunately, only one patient had a VUR (i.e., grade 1). While VUR can be a complication after a prolonged period of high intravesical pressure in individuals with NLUTD [33], recent studies reported a similar low prevalence of VUR in individuals with acute SCI (i.e., within the first year after injury) [27,34]. Given the paucity of longitudinal urodynamic data in patients with MCS/UWS, at the present time, one can only refer to other entities of NLUTD, such as SCI and MS, regarding the risk of developing VUR long-term. While Sirasaporn and Saengsuwan [35] reported an overall low incidence (i.e., 7.5 cases per 100 person-years) in a historical study comprising individuals with chronic SCI, Piquet et al. [36] also reported a low prevalence (i.e., 5.5%) in individuals with MS.

While the prevalence of a low-compliance bladder was approximately one in six in patients with SCI (i.e., 17%) [37], we found that one in four patients in our cohort had a low-compliance bladder. This surprising finding emphasizes the need for conducting VUDS early in patients with MCS/UWS, as half the patients in the “Late VUDS” group presented with a compliance of less than 20 cmH_2_O compared to one in five patients in the “Early VUDS” group.

### 4.3. LUT Management before and after VUDS

Prior to conducting VUDS, most patients (i.e., >75%) utilized assisted urinary bladder emptying by means of indwelling catheterization. To avoid catheter-associated complications such as CA-UTI, one should refrain from using indwelling catheters to manage urinary incontinence whenever possible [26].

VUDS can help to identify potential patients eligible to switch bladder emptying methods, i.e., condom catheter or intermittent catheterization, as alternatives to indwelling catheters. A variety of indwelling catheter-associated complications, such as CA-UTI, urethral trauma, urethritis, fistula, bladder neck incompetence, sphincter erosion, bladder stones, bladder cancer, and allergies, have been described extensively [38]. We recorded complications in about half of our cohort (i.e., 47%), mostly associated with indwelling catheterization, of which a third were UTIs/CAUTIs [26]. Although no data from randomized controlled trials on bladder emptying methods in patients with MCS/UWS are available, previous studies showed the benefit of intermittent catheterization or urinal condoms vs. indwelling catheters in patients with NLUTD [39].

Prior to VUDS, more than one in five patients received voiding-altering medications. There is evidence that routine administration of anticholinergic drugs may benefit patients with MCS/UWS. Krimanchsky et al. showed that while decreasing detrusor pressure and increasing bladder capacity with an anticholinergic agent, external signs of pain on emptying disappeared [12]. Considering our high number of NDOs with considerably high intravesical pressure, routine application of anticholinergic agents could be considered.

Following VUDS, LUT management with respect to bladder emptying methods was changed in half of our cohort. In patients with sufficient reflex voiding, condom catheters were used to successfully eliminate indwelling catheters and avoid urinary incontinence. Intermittent catheterization was not performed in our cohort and appears to be not a practical option in MCS/UWS patients due to the limited level of consciousness.

On account of our urodynamic findings, we commenced pharmacological treatment in 75% of patients to aid their NLUTD, mainly antimuscarinics and alpha-blockers. We recommended intradetrusor botulinum toxin-A injections in two cases to lower elevated detrusor pressure and intrasphincteric botulinum toxin-A injections in six patients to reduce subvesical resistance and allow condom catheters to collect urine without endangering the UUT, respectively. With respect to the aforementioned literature in patients with MCS/UWS [12,13,14,15,16], only Girando et al. [13] reported on the effect of pharmacological treatment on urodynamic parameters (i.e., before and after continuous intrathecal baclofen, ITB). While MCC and PVR urine were increased after ITB, there was a smaller number of patients with NDO and a lower detrusor leak point pressure [13].

The best time point for using VUDS to investigate LUT function in patients with MCS/UWS has yet to be determined. Jiang et al. [40] showed in an experimental trial in rats that there was a transient reduction in bladder contractility immediately after traumatic brain injury (TBI). However, one month after TBI, they showed an increase in overactive bladder or urinary urge incontinence [40]. This suggests that urodynamic evaluation/VUDS should not only be performed during the acute phase of TBI but repeated 4–6 weeks after TBI to surpass the initial shock phase in order to avoid relying on false-negative findings from the very first assessment after an acquired brain injury [40]. Furthermore, regular follow-up urodynamic evaluations/VUDS should be conducted to monitor LUT function, detect its decline, and adapt LUT management accordingly in MCS/UWS patients, as recommended in other patient groups with NLUTD [41].

### 4.4. Limitations

The retrospective nature of our study and the lack of performing VUDS in all MCS/UWS patients systematically at our institute led to a selection bias. In the past, whether to perform VUDS and when was not predetermined. A number of VUDS have been performed because of underlying urological complications, leading to a selection bias, i.e., patients with apparent LUT dysfunction. This might contribute to the high prevalence in three of the five unfavorable urodynamic parameters [20].

Furthermore, after adjusting LUT management, we did not perform follow-up VUDS systematically. Thus, we cannot report on the long-term efficacy of the adjusted LUT management. Despite the recommendation of long-term UDS surveillance in other entities with NLUTD [19], the value of follow-up VUDS in patients with UWS/MCS still needs to be determined. To answer this, our institute has started to systematically perform UDS at the following time points: (1) between 2 and 6 months, (2) at 12 months, and (3) at 24 months following a brain injury/diagnosis of MCS/UWS. Depending on the extent of NLUTD, additional VUDS will be performed individually to monitor LUT function as well as treatment efficacy.

## 5. Conclusions

This retrospective cohort study revealed a high prevalence of unfavorable urodynamic findings in patients with MCS/UWS, which was very similar to individuals with SCI. These findings illustrate the importance of early VUDS in order to adapt LUT management in this cohort accordingly. Considering the paucity of evidence in this cohort, further research is needed to track how urinary tract function is developing long-term and whether additional unfavorable urodynamic parameters will arise in the course of the disease while patients are aging. Lastly, given the aforementioned, clinicians taking care of patients with MCS/UWS need to strive for the implementation of follow-up strategies rigorously to minimize the risk of deterioration of the entire urinary tract function long-term.

## Figures and Tables

**Figure 1 biomedicines-11-02432-f001:**
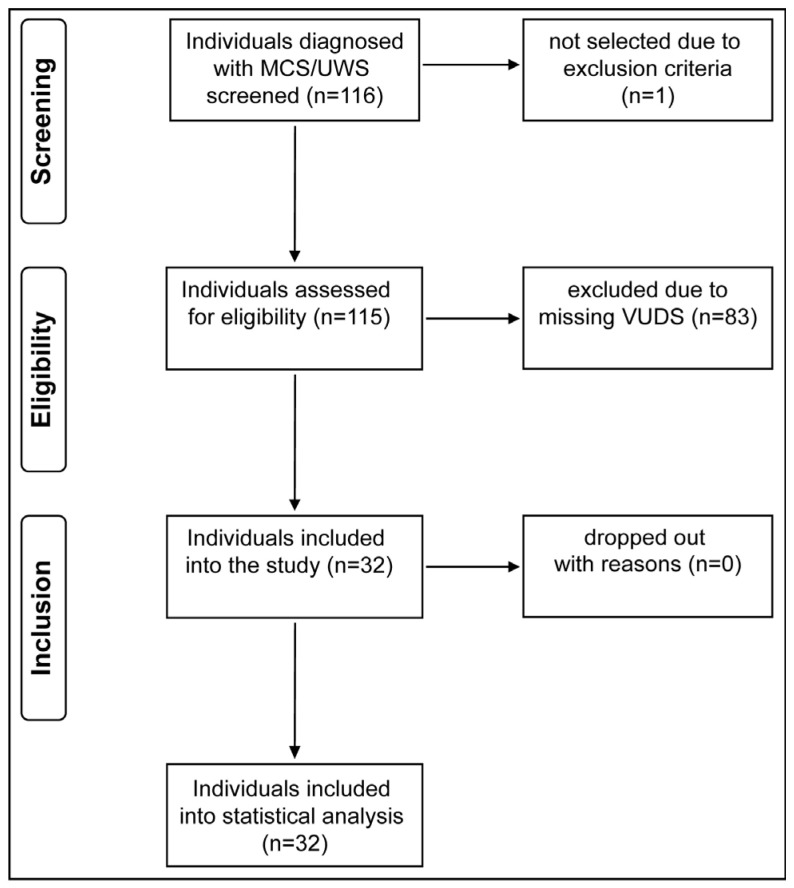
Study flow diagram. From a pool of 116 screened individuals, 32 fulfilled all inclusion criteria and were included in our analysis. MCS/UWS = minimally conscious state/unresponsive wakefulness syndrome, VUDS = video-urodynamic studies.

**Table 1 biomedicines-11-02432-t001:** Patient demographics and injury characteristics.

Demographics	All Patients (n = 32)	Early VUDS (n = 26)	Late VUDS (n = 6)	*p* Value ^$^
Age [years]	37 [28; 52], 20–74	37 [28; 51], 20–69	36 [28; 58], 23–74	1.0
Time from diagnosis to VUDS [months]	4 [2; 6], 1–93	2.5 [2; 4], 1–7	32 [28; 74], 25–93	**0.03**
Sex (female/male)	7 (21.9%)/25 (78.1%)	7 (26.9%)/19 (73.1%)	0/6	0.3
**Injury characteristics**				
Type of brain injury				0.8
*Traumatic*	*18* (*56.3%*)	*15* (*57.7%*)	*3* (*50%*)	
*Hypoxic*	*13* (*40.6%*)	*10* (*38.5%*)	*3* (*50%*)	
*Hemorrhagic*	*1* (*3.1%*)	*1* (*3.8%*)	*0*	
Coma Recovery Scale-revised (n = 19) *	16 [7.5; 21.5], 3–24	9 [6; 22.2], 3–24	15 [10; 16], 8–17	0.1
Glasgow Coma Scale (n = 26) *	9.5 [6.2; 10.8], 3–15	8 [6; 10.8], 3–15	10 [9.2; 10], 8–11	0.3
Bavesta Score (n = 19) *	2.8 [1.5; 3.6], 0.8–5.0	3 [1.5; 3.7], 0.8–5.0	2.2 [1.6; 3.1], 1.2–3.6	0.2

Results are shown as median, lower and upper quartiles, and range for age, Coma Recovery Scale-revised, Glasgow Coma Scale, and Bavesta Score. * Indicates the number of patients with complete data. ^$^ indicates the comparison between early and late VUDS. *p* value in bold highlights a statistically significant between-group (i.e., “Early VUDS” vs. “Late VDUS”) difference.

**Table 2 biomedicines-11-02432-t002:** Lower urinary tract management before and after baseline urodynamic studies.

Pharmaceutical LUT Treatment	All Patients (n = 32)		Early VUDS (n = 26)		Late VUDS (n = 6)	
	Before	After	Before	After	Before	After
No. of patients on LUT treatment	7 (21.9%)	24 (75%) *	7 (26.9%)	19 (73.1%) **	0	5 (83.3%) ***
*Antimuscarinics*	*4*	*10*	*4*	*8*	*0*	*2*
*Alpha-blockers*	*3*	*11*	*3*	*8*	*0*	*3*
*Mirabegron*	*0*	*1*	*0*	*1*	*0*	*0*
*Intradetrusor botulinum toxin-A*	*0*	*2*	*0*	*2*	*0*	*0*
*Intrasphincteric botulinum toxin-A*	*0*	*6*	*0*	*3*	*0*	*3*
**Bladder-emptying methods**						
Indwelling catheters	25 (78.1%)	12 (37.5%)	21 (80.8%)	10 (38.5%)	4 (66.7%)	2 (33.3%)
*Transurethral*	*7*	*2*	*7*	*2*	*0*	*0*
*Suprapubic*	*18*	*10*	*14*	*8*	*4*	*2*
Intermittent catheterization	0	0	0	0	0	0
NDO incontinence	6 (18.8%)	19 (59.4%)	4 (15.4%)	15 (57.7%)	2 (33.3%)	4 (66.7%)
*with condom catheter*	*5*	*18*	*0*	*14*	*2*	*4*
*without condom catheter*	*1*	*1*	*0*	*1*	*0*	*0*
Volitional voiding	1 (3.1%)	1 (3.1%)	1 (3.8%)	1 (3.8%)	0	0

LUT = lower urinary tract. NDO = neurogenic detrusor overactivity. Patients who received two LUT treatment options, i.e., * n = 6, ** n = 3, *** n = 3.

**Table 3 biomedicines-11-02432-t003:** Video-urodynamic parameters and distribution of unfavorable findings.

Video-Urodynamic Parameters	All Patients (n = 32)	Early VUDS (n = 26)	Late VUDS (n = 6)	*p* Value ^$^
NDO * [yes/no]	25 (78.1%)/7 (21.9%)	19 (78.1%)/7 (26.9%)	6/0	0.3
Pdetmax storage [cmH_2_O]	52 [21; 76], 5–272	49 [15; 62], 5–190	91 [81; 186], 51–272	0.3
Pdetmax storage > 40 cmH_2_O * [yes or no]	21 (65.6%)/11 (34.4%)	15 (57.7%)/11 (42.3%)	6/0	0.07
DOLPP [cmH_2_O]	42 [33; 52], 12–95	39 [30; 51], 12–95	43 [42; 61], 38–90	0.6
DOLV [mL]	98 [22; 241], 0–450	78 [0; 219], 0–450	185 [88; 290], 65–320	0.6
Compliance [mL/H_2_O]	29 [21; 44], 4–116	36 [24; 45], 5–116	18 [12; 28], 4–55	0.8
Low-compliance bladder * [yes or no]	8 (25.0%)/24 (75.0%)	5 (19.2%)/21 (80.8%)	3 (50.0%)/3 (50.0%)	
MCC [mL]	238 [140; 351], 60–600	238 [150; 359], 60–600	215 [95; 331], 65–350	0.8
PVR urine [yes/no]	20 (62.5%)/12 (37.5%)	16 (61.5%)/10 (38.5%)	4 (66.7%)/2 (33.3%)	
PVR urine [mL]	35 [0; 116], 0–530	45 [0; 120], 0–530	30 [8; 30], 0–45	0.09
DSD * [yes/no]	22 (68.8%)/10 (31.2%)	17 (65.4%)/9 (34.6%)	5 (83.3%)/1 (16.7%)	0.6
VUR * [yes/no]	1 (3.1%)/31 (96.9%)	0/26	1 (16.7%)/5 (83.3%)	-

DOLPP = detrusor overactivity leak point pressure, DOLV = detrusor overactivity leak volume, DSD = detrusor sphincter dyssynergia, MCC = maximal cystometric capacity, NDO = neurogenic detrusor overactivity, Pdetmax = maximal detrusor pressure, PVR = post-void residual, VUR = vesico–uretero–renal reflux (i.e., grade 1). * Indicates unfavorable urodynamic findings. ^$^ Indicates the comparison between early and late VUDS.

## Data Availability

De-identified data are available upon request from the authors.

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
