# Peer review of "Prevalence of Unfavorable Video-Urodynamic Findings and Clinical Implications in Patients with Minimally Conscious State/Unresponsive Wakefulness Syndrome: A Retrospective Descriptive Analysis"

_biomedicines, 2023, doi:10.3390/biomedicines11092432_

Round 1

Reviewer 1 Report

Dear authors,

This is a very interesting and absolutely scientific work in very selected Neuro-urological population. 

However, there are some major concerns to be clarified.

- there is no clear documentation of the time relapsed from the time of the neurological disorder till VUDS performance. This could be crucial for detrusor respond, especially during the filling phase and more specifically in patients with an indexing catheter. It could be preferable, if you perform a separate statistical analysis, including those patients and the time since catheter has been placed.

- in cases that there is an existing pharmaceutical treatment, it is useful to mention if you have asked for anticholinergics or a-blockers to be stopped and for how many days before the VUDS

- please, provide information about bladder compliance in your results and tables

Author Response

Response to reviewer 1

We thank reviewer 1 for their valuable feedback on our manuscript. We have included a marked version of our revised manuscript with these changes highlighted in yellow. Please find our point-by-point responses to each comment below.

Reviewer 1:

Dear authors,

This is a very interesting and absolutely scientific work in very selected Neuro-urological population. However, there are some major concerns to be clarified.

- there is no clear documentation of the time relapsed from the time of the neurological disorder till VUDS performance. This could be crucial for detrusor respond, especially during the filling phase and more specifically in patients with an indexing catheter. It could be preferable, if you perform a separate statistical analysis, including those patients and the time since catheter has been placed.

Authors’ response:

We thank the reviewer for their comment. We have previously stated in line 132 that the median time from diagnosis to VUDS was 4 months. However, we agree with the reviewer that it would be better to provide further information on that matter. We have added further details (lower and upper quartiles, minimum and maximum) in Table 1.

Twenty-six of 32 patients underwent VUDS within the first 7 months (i.e., ‘Early VUDS’) following diagnosis, i.e. 2.5 [2 ; 4], 1 – 7. The remaining six patients underwent VUDS significantly later (i.e. ‘Late VUDS’) that is 25, 28, 30, 34, 88, and 93 months following diagnosis, respectively, i.e. 32 [28 ; 74], 25 – 93.

Thus, we dichotomized the entire cohort into ‘Early VUDS’ and ‘Late VUDS’ groups (i.e., Table 1 to 3) and compared both accordingly (i.e., Table 1 and 3). However, we did not find another statistically significant between-group differences (except to from diagnosis to VUDS).

- in cases that there is an existing pharmaceutical treatment, it is useful to mention if you have asked for anticholinergics or a-blockers to be stopped and for how many days before the VUDS

Authors’ response:

We thank the reviewer for their comment. However, we did not stop any treatment prior to VUDS as wanted to evaluate whether pre-existing therapy was sufficient or not. However, only seven (22 %) out of 32 patients were on either antimuscarinics (n = 4) or alpha-blockers (n = 3) prior to VUDS, while the remaining patients (78 %, 25/32) were pharmacologically naïve with respect to LUT treatment.

- please, provide information about bladder compliance in your results and tables

Authors’ response:

We thank the reviewer for their comment and agree the usefulness of proving these data. Thus, we computed bladder compliance and included the results within table 3. Furthermore, we commented on these results in a designates paragraph at the end of the discussion section.

Reviewer 2 Report

Thank you for submitting your interesting paper!

First of all, I suggest you clearly define "unfavorable findings", starting with the abstract and more detailed in the body of the text. It improves readability.

While I understand the retrospective nature of your research, it would be interesting to know which was the indication for VUDS, since it was not been done routinely in this population.

It is not clear for me whether the patients undergoing urodynamics which  were already on some treatment were analyzed together with the ones (the majority according to your results) who only started treatment for LUTS after the investigation.

For me, the real utility of UDS is not to identify abnormalities (which are highly likely in your population) but to identify useful, effective treatments, and we know this population has poor response and compliance to medical treatments.

I kindly ask you to have a look and cite the following paper, on a very similar topic: doi: 10.5173/ceju.2014.03.art12

You advocate the use of CIC for your population, I would like to see a short line about their possibility to actually perform it and the likelihood this will happen.

Author Response

Response to reviewer 2

We thank reviewer 2 for their valuable feedback on our manuscript. We have included a marked version of our revised manuscript with these changes highlighted in yellow. Please find our point-by-point responses to each comment below.

Reviewer 2:

Thank you for submitting your interesting paper!

First of all, I suggest you clearly define "unfavorable findings", starting with the abstract and more detailed in the body of the text. It improves readability.

Authors’ response:

We thank the reviewer for their comment. We have previously referred to "unfavorable findings" in lines 80 to 83 as ‘high maximum detrusor pressure [Pdet] during the storage phase [> 40 cmH2O], neurogenic detrusor overactivity (NDO), detrusor sphincter dyssynergia (DSD), and VUR’ – in accordance to Welk B et al. Early urological care of patients with spinal cord injury. World J Urol. 2018 Oct;36(10):1537-1544. doi: 10.1007/s00345-018-2367-, which was cited as Reference # 20. However, we have revised this statement in accordance with your comment and the ones from the other reviewers as follows:

‘The primary outcome was to obtain the prevalence of unfavorable urodynamic findings [20] (i.e., high maximum detrusor pressure [Pdet] during the storage phase [> 40 cmH2O], low-compliance bladder [< 20 mL/cm H2O], neurogenic detrusor overactivity (NDO), detrusor sphincter dyssynergia (DSD), and VUR) during early VUDS in patients with MCS/UWS.’

Furthermore, as suggested by the reviewer, we have revised the abstract accordingly to explain what "unfavorable VUDS findings" actually are.

While I understand the retrospective nature of your research, it would be interesting to know which was the indication for VUDS, since it was not been done routinely in this population.

Authors’ response:

We thank the reviewer for their comment. We initiated to perform VUDS, since a significant number of patients presented with urological complications, such as reflex incontinence, vegetative stress, pain despite continuous bladder drainage using indwelling catheter including pseudo-occlusions, high rates of urinary tract infections or sediment discharge. Furthermore, few patients went into a severe septic shock. With respect to our knowledge from individual with spinal cord injury (SCI) or multiple sclerosis (and in line with the European Urology guidelines on Neuro-Urology, we conducted VUDS to assess the extent as well as progress of NLUTD in order to adapt bladder / LUT  treatment accordingly.

It is not clear for me whether the patients undergoing urodynamics which were already on some treatment were analyzed together with the ones (the majority according to your results) who only started treatment for LUTS after the investigation.

Authors’ response:

We thank the reviewer for their comment. Due to the relatively small number of examinations, we decided against making a distinction. In everyday clinical practice, we try to always conduct VUDS to establish a baseline (i.e., extent of NLUTD) before starting appropriate therapy (similar to individuals with SCI), if it was justifiable. However, this was not always possible in the past. Also, it was usually ethically and medically not justifiable to pause medication before VUDS that were conducted later. Performing VUDS in severely brain-injured patients with, for example, recurrent infections, ventilation, delirium, agitation, vegetative stress, spasticity is often delayed and sometimes, therapy must be started prior to baseline VUDS in order to avert further dangers from the patient.

For me, the real utility of UDS is not to identify abnormalities (which are highly likely in your population) but to identify useful, effective treatments, and we know this population has poor response and compliance to medical treatments.

Authors’ response:

We thank the reviewer for their comment. In fact, the main purpose of VUDS is to assess the extent of NLUTD (at baseline, similar to patient following acute SCI but after the initial spinal shock phase) and to follow up patient that are on treatment to assess treatment efficacy, especially in out-patients. From our experience, taking care of patients with MCS/UWS, it is often difficult to choose an appropriate therapy option or measure therapy efficacy, because potential side-effects form antimuscarinics or contraindications due to other medications (antiepileptics), the type of application (non-probe preparations) or the accessibility of the special consultation when patients are transitioning from in- to out-patients. It is therefore important to have therapy concepts that are easy to apply and sustainable in everyday life.

I kindly ask you to have a look and cite the following paper, on a very similar topic: doi: 10.5173/ceju.2014.03.art12

Authors’ response:

We thank the reviewer for their comment. We read the suggested systematic review. While, the topic concerns NLUTD, it does not include or discuss patients with UWS / MCS. Thus, we feel that the suggested work does not fit. Therefore, we did not include the suggested work as a reference. If the reviewer can provide a more fitting reference concerning patients with UWS / MCS, the author would add this gladly.

You advocate the use of CIC for your population, I would like to see a short line about their possibility to actually perform it and the likelihood this will happen.

Authors’ response:

We thank the reviewer for their comment. Intermittent catheterization (IC) was not performed in our cohort – as mentioned in line 337-338 (‘Intermittent catheterization was not performed in our cohort and appears to be not a practical option in MCS/UWS patients due to the limited level of consciousness.’) – but was highlighted as a beneficial bladder emptying method in patient with NLUTD in lines 223-226 (‘Although no data from randomized controlled trials on bladder emptying methods in patients MCS/UWS are available, previous studies showed a benefit of intermittent catheterization or urinal condom vs. indwelling catheters in patients with NLUTD [38]’).

Indeed, IC would only be a valid option for patients with MCS/UWS if one could achieve competent attenuation of NDO and ensure that IC is performed “assisted” regularly by nurses or other caretakers, such as partners or family members. However, this is not often possible.

Reviewer 3 Report

an interesting manuscript on urination dysfunction in a very selective group of patients - MCS/UWS -  one of the first reports, according to authors, using VUDS in this group of patients

several major findings are observed by the authors - very high frequency of unfavorable VUDS findings, and the need for a change of urinary tract management in over 3/4 of the studied patients.

only one minor technical fault - row 152 - the abbreviation should be NDU, not repetition of NDO

Author Response

Response to reviewer 3

We thank reviewer 3 for their kind feedback on our manuscript. We have included a marked version of our revised manuscript with these changes highlighted in yellow. Please find our point-by-point responses to each comment below.

Reviewer 3:

an interesting manuscript on urination dysfunction in a very selective group of patients - MCS/UWS -  one of the first reports, according to authors, using VUDS in this group of patients

several major findings are observed by the authors - very high frequency of unfavorable VUDS findings, and the need for a change of urinary tract management in over 3/4 of the studied patients.

only one minor technical fault - row 152 - the abbreviation should be NDU, not repetition of NDO

Authors’ response:

We thank the reviewer for the kinds words. We apologize for our oversight. After review of the data, we revised the manuscript by removing the data on NDU as it would indicate a reduced contractility during an attempt to void. However, only one individual was able to void volitionally.

Round 2

Reviewer 1 Report

Accepted with the revised corrections.

Reviewer 2 Report

Thank you for making the suggested changes.